# Risk factors associated with Hepatitis B virus infection among pregnant women attending public hospitals in Addis Ababa, Ethiopia

**Mebrihit Arefaine Tesfu** *, **Tilahun Teklehaymanot Habtemariam, Nega Berhe Belay**

Aklilu Lemma Institute of pathobiology, Addis Ababa University, Addis Ababa, Ethiopia

* mebrihitarefaine@gmail.com

## Abstract

### Background

Hepatitis B Virus (HBV) infection is one of the serious public health problems worldwide and is a major cause of morbidity and mortality. Viral hepatitis during pregnancy poses problems like a high risk of maternal complications, mother-to-child transmission (MTCT), and challenges in the management of drugs. This study aimed to determine the magnitude of HBV infection and associated risk factors among pregnant women who attended public hospitals in Addis Ababa, Ethiopia.

### Method

A multicenter prospective cohort study with a nested case-control was conducted from January 2019 to December 2020 in 5 public hospitals with maternal and child health care services in Addis Ababa. Three hundred pregnant women whose screening results for Hepatitis B surface antigen (HBsAg) were positive and another 300 with negative HBsAg were involved. Laboratory test results of blood samples and structured questionnaires were used to collect the data. Data was entered and analyzed by SPSS version 20 software using descriptive and logistic regression analyses.

### Results

Of the 12,138 pregnant women who screened for HBsAg as routine antenatal care (ANC), 369 (3.04%) were positive. All of the sociodemographic characteristics did not significantly differ in both the cases and the controls. Body tattooing (AOR = 1.66; 95 CI: 1.008–2.728), multiple sexual partners (AOR = 2.5; 95% CI: 1.604–3.901), family history of HBV (AOR = 2.62; 95% CI: 1.239–5.547), and sharing sharp materials (AOR = 3.02; 95% CI: 1.87–4.87) were factors associated with increased risk of HBV infection.

### Conclusions

An intermediate endemicity of HBV infection was detected among pregnant women. Body tattooing, having multiple sexual partners, family history of HBV, and sharing sharp materials were significantly associated with HBV infection. Awareness creation on the mode of

**Data Availability Statement:** All relevant data are within the manuscript.

**Funding:** The study's fund was obtained from Addis Ababa University, Ethiopia. The funder had

no role in designing the study, data collection, analysis, or manuscript preparation.

**Competing interests:** The authors have declared that no competing interests exist.

transmission and early screening of all pregnant women for HBsAg must be strengthened to minimize and control the spread of the infection.

## Introduction

HBV infection is one of the serious public health problems worldwide and is a major cause of morbidity and mortality [1]. HBV is the cause of; infection for more than 2 billion people, 350 million chronic carriers, and almost 1 million annual deaths due to related liver complications globally [2]. In Africa, there are about 65 million individuals who carry HBV, with a 25% mortality risk. In sub-Saharan Africa, the prevalence of HBV infection ranges between 9–20% [3]. Viral hepatitis during pregnancy poses problems like a high risk of maternal complications, a high rate of MTCT, and challenges in the management of drugs [4, 5].

Transmission of HBV occurs with percutaneous and per mucosal exposure to infectious body fluids from persons with acute or chronic HBV infection [6]. The highest HBV concentration occurs in the blood and other body fluids like semen and vaginal secretions [6, 7]. Thus, several risk factors are predictors of HBV infection including, blood transfusion, from mother to child, sharing needles and razor blades, undergoing surgical, dental, medical, or delivery procedures, unprotected sex, history of sexually transmitted infections, and history of abortion [5, 7–10]. Perinatal and early childhood transmissions are the main routes of HBV infection in endemic areas [11, 12]. The risk of MTCT of HBV is related to the HBV replicative status of the mother, which correlates with the presence of Hepatitis B e antigen (HBeAg) and high HBV Deoxyribonucleic acid (DNA) levels [13]. HBV causes chronic hepatitis B infection in up to 90% of exposed infants who do not receive appropriate immunoprophylaxis in contrast to 10–25% of children and 5–10% of exposed immunocompetent adults [4, 12]. Furthermore, the infected offsprings become reservoirs for subsequent horizontal infection [14]. Periodic perinatal HBV screening, administration of immunoprophylaxis with both Hepatitis B immunoglobulin (HBIg) and vaccine for the newborn of HBV-infected mothers, and vaccination of high-risk mothers and children reduce the risk of HBV transmission [12, 13].

A meta-analysis and systematic review conducted in Ethiopia showed 6.3% of overall HBsAg prevalence in the general population over the last five decades [15]. HBV infection prevalence rate of 1.2% in Iran [16], 9.2% in the Gambia [17], 7.7% in Ghana [18], 4.12% in Somali [10], 3.2% in Eritrea [19], and 1.94% - 11.6% in Ethiopia [20–27] were reported in pregnant women.

In Ethiopia, even if there is little published data on the prevalence of HBV and associated risk factors among pregnant women, to the best of our knowledge, there is no case-control study that helps to better investigate these risk factors. Hence this study aimed to determine the magnitude of HBV infection and associated risk factors among pregnant women who attended public hospitals in Addis Ababa, Ethiopia. The findings of this study will be used to design appropriate prevention and control measures and intervention strategies against the transmission of HBV infection.

## Methods and materials

### Study design, period, site, and population

A multicenter prospective cohort study with a nested case-control was conducted from January 2019 to December 2020 in Addis Ababa, the capital city of Ethiopia, with an estimated population of 4,591,983 in 2019 [28]. It was conducted in 5 public hospitals with maternal and

child health care services in the city. The hospitals were Gandhi Memorial Hospital, a regional referral that provides gynecologic and maternity services and daily managed 25–40 deliveries for mothers from Addis Ababa and its surroundings. Zewditu Memorial Hospital which is Ethiopia's leading hospital in the treatment of Antiretroviral therapy (ART) patients and also gives maternal and child health care services, Yekatit 12 Hospital Medical College provides services for more than 500,000 people living in its catchment area, Menilik II referral Hospital which is the first hospital in Ethiopia serving the society for years especially known for tertiary eye care and forensic medicine, and Armed Force Referral and Teaching Hospital, which provides medical services to members of the Ethiopian defense forces. The study population was pregnant women who attended ANC or delivery services in the selected public hospitals in Addis Ababa.

## Sample size determination and sampling technique

To determine the MTCT of HBV and the protective effect of the immune-prophylaxis vaccine in Ethiopia, a sample size of 369 HBV-positive pregnant women was calculated using, 40% of MTCT of HBV in the absence of post-exposure prophylaxis in Nigeria [29], and giving any particular outcome to be with 5% marginal error, 95% confidence interval, and using the following single population determination formula.

$$n = \frac{(z\,\partial/2)^2 p(1-p)}{d^2}$$

To get 369 HBsAg positive, 12,318 pregnant women were screened for HBsAg as routine ANC service on their first visit in the selected public hospitals during the study period, and 300 of them were included in this study. The women whose screening results were positive for HBsAg were considered as cases. The purposive sampling technique was used to get the cases, and, for each HBsAg positive woman, the next HBsAg negative woman was interviewed and acted as a control. Thus 300 cases and 300 controls were involved in the study.

## Eligibility

**Inclusion criteria.** All pregnant women who attended antenatal care or delivery services in the selected hospitals in Addis Ababa and those who volunteered to participate and give informed consent were included.

**Exclusion criteria.** All pregnant women who were seropositive for HIV were excluded.

## Data collection tools and procedures

Pre-designed and pre-tested structured questionnaires were used to get information on socio-demographic characteristics and associated risk factors with HBV infection. Ten midwives and nurses blinded to the HBsAg status of the participants, who worked in the study hospitals, collected the data. Two days of training were given on the objective of the study, obtaining consent, the confidentiality of the information, and data collection procedures for the data collectors. Questionnaires were carefully designed and pre-tested with individuals equivalent to 5% of the calculated sample size in Ras Desta Damtew Hospital and were slightly amended after pre-tested results revealed a lack of clarity for some responses. Five ml of venous blood was collected based on the standard collection procedure to determine the HBsAg status of the pregnant women as routine ANC service in their first ANC visit. The HIV serostatus of the pregnant woman was determined by reviewing their ANC records after their second ANC

visit and above. The supervisors and the principal investigator supervised the data collection process.

**Laboratory procedures.** The blood samples were centrifuged at 3000 revolutions per minute (RPM) for at least 10 minutes at room temperature and tested for HBsAg using rapid test Cassettes (Nantong diagnosis biotechnology co. Ltd P. R. china), which have specificity and sensitivity of greater than 99% following the manufacturer's protocol.

## Operational definitions

Case: A pregnant woman who screened positive for HBsAg.
Control: A pregnant woman who screened negative for HBsAg.

## Data management and analysis

The generated data was cleaned, coded, and uploaded into a computer using Statistical Package for the Social Sciences (SPSS) version 20.0 statistical software for analysis and interpretation. Descriptive values were expressed as the frequency, percentage, and mean ± standard deviation (SD). Logistic regression analysis was implemented to explore and determine the relationship of predictors to outcome variables. Variables significant at $p < 0.25$ with the dependent variable in univariate analysis were selected for multivariable analysis. An odds ratio with a 95% confidence level was computed, and a significant association was declared at $p < 0.05$.

## Ethical consideration

Ethical clearance and approval were obtained from the Institutional Review Boards of Aklilu Lemma Institute of Pathobiology, Addis Ababa University, and the Addis Ababa city administration health bureau. Permission to carry out the study was obtained from the selected public hospitals, and after explaining the purpose, written informed consent was obtained from the pregnant women. Moreover, confidentiality was assured for all the information provided, and the personal identifiers were not included in the questionnaires. All methods were carried out following relevant guidelines and regulations.

## Results

### Socio-demographic characteristics of the study participants

Twelve thousand one hundred thirty-eight (12,138) pregnant women were screened for HBsAg as routine antenatal care in the selected public hospitals during the study period, and 369 (3.04%) were positive for HBsAg. Of the HBsAg-positive pregnant woman, 60 (16.3%) refused the interview, and 9 (2.4%) were HIV seropositive. Those 300 HBsAg—positive pregnant women and another 300 HBsAg- negative ones who served as control formed the basis of further analysis. The mean age of cases and controls was 27.78 ± 4.74 and 28.28± 4.78 years, respectively. The majority of the cases (42.7%) and the controls (41.7%) were in the age group of 25–29 years. Most (91.7%) of the cases and 92.7% of the controls were married. All of the sociodemographic characteristics did not significantly differ in both the cases and the controls (Table 1).

### Risk factors associated with HBV infection among pregnant women

Differences in some risk factors of HBV infection were evident between cases and controls. Having a history of blood transfusion (7.3% vs.3.7%), sexually transmitted diseases (STDs) (7.7% vs. 3%), body tattooing (20.7% vs. 11.3%), having multiple sexual partners (29.7% vs.

**Table 1. Socio-demographic characteristics of pregnant women attending public hospitals in Addis Ababa.**

| Variable | Category | Cases (n = 300) N (%) | Controls (n = 300) N (%) | COR (95%CI) | P value |
|---|---|---|---|---|---|
| Age | <20 | 7 (2.3) | 6 (2) | 0.8(0.24–2.67) | 0.717 |
| | 20–24 | 72 (24) | 62 (20.7) | 0.804(.434–1.49) | 0.488 |
| | 25–29 | 128 (42.7) | 125 (41.7) | 0.911(.515–1.613) | 0.750 |
| | 30–34 | 65 (21.7) | 77 (25.7) | 1.106(.600–2.038) | 0.748 |
| | ≥35 | 28 (9.3) | 30 (10) | ref | |
| Marital status | single | 15 (5) | 16 (5.3) | 1.78 (.518–6.101) | 0.360 |
| | Married | 275 (91.7) | 278 (92.7) | 1.69 (.604–4.699) | 0.319 |
| | Others* | 10 (3.33) | 6 (2) | Ref | |
| Educational status | Illiterate | 32 (10.7) | 27 (9) | 0.63 (.35–1.14) | 0.124 |
| | Primary level | 104 (34.7) | 81 (27) | .58 (.383–.879) | 0.10 |
| | Secondary level | 88 (29.3) | 90 (30) | .762 (.502–1.16) | 0.202 |
| | College diploma and above | 76 (25.3) | 102 (34) | Ref | |
| Religion | Orthodox Christian | 213 (71) | 200 (66.7) | .939 (.298–2.96) | 0.914 |
| | Muslim | 50 (16.7) | 50 (16.7) | 1.00 (.302–3.312) | 1.000 |
| | Protestant | 31 (10.3) | 44 (14.7) | 1.42 (.418–4.814) | .574 |
| | Others ◇ | 6 (2) | 6 (2) | Ref | |

COR: crude odds ratio, CI: confidence interval

* includes widowed &divorced

◇ includes catholic and waqoyofeta

13%), sharing sharp materials (31% vs. 10.7%), family history of HBV (12% vs. 4%) and, history of jaundice (6% vs. 1.3%) were more frequent in cases than controls. Other risk factors like history of abortion, surgical procedure, dental procedure, ear/nose piercing, female circumcision, hospital admission, and history of contact with jaundice patients presented similar frequencies in both cases and the controls (Table 2).

In univariate analysis, cases were more likely to report a history of STDs (OR = 2.69; 95% CI: 1.22–5.90), body tattooing (OR = 2.04; 95 CI: 1.295–3.21) multiple sexual partners (OR = 2.82; 95% CI: 1.86–4.29), family history of HBV (OR = 3.27; 95% CI: 1.67–6.42), sharing sharp materials (OR = 3.76; 95% CI: 2.42–5.85), and history of jaundice (OR = 4.72; 95% CI:1.58–14.13) than controls.

In multivariate analysis, body tattooing (AOR = 1.66; 95 CI: 1.008–2.728), multiple sexual partners (AOR = 2.5; 95% CI: 1.604–3.901), family history of HBV (AOR = 2.62; 95% CI: 1.239–5.547), and sharing sharp materials (AOR = 3.02; 95% CI: 1.87–4.87) were factors significantly associated with increased risk of HBV infection (Table 2).

## Discussion

The seroprevalence of HBsAg in this study was 3.04% and indicated intermediate endemicity of HBV infection according to WHO classification criteria. This finding is similar to studies reported in Addis Ababa (3%) [30] and Jimma (3.7%) [31]. But it is higher than prevalence studies in the East Wollega zone (2.4%) [21] and Bahirdar (1.94%) [20]. However, it is lower than studies from Arba Minch (4.3%) [22], Harar city (6.3%) [23], Deder Hospital (6.9%) [24], Hawassa referral Hospital (7.8%) [25], Gambella (7.9%) [26], and in Tigray region (11.6%) [27]. On the other hand, in comparison with other countries, higher results were reported in Gambia (9.2%) [17], Sudan (5.6%) [32], Ghana (7.7%) [18], and Somalia (4.12%) [10]. This variation might be due to differences in geographical, cultural practices, socioeconomic,

**Table 2. Risk factors associated with HBV infection among pregnant women attending public hospitals in Addis Ababa.**

| variables | category | cases (n = 300) n (%) | Controls (n = 300) n (%) | Bivariate analysis | | multivariate analysis | |
|---|---|---|---|---|---|---|---|
| | | | | P value | COR (95%CI) | AOR (95%CI) | P value |
| History of blood transfusion | Yes | 22 (7.3) | 11 (3.7) | 0.053 | 2.08 (.99–4.37) | 1.65 (.706–3.876) | 0.247 |
| | No | 278 (92.7) | 289 (96.3) | Ref | | | |
| History of STDs | Yes | 23 (7.7) | 9 (3) | 0.014 | 2.69 (1.22–5.90) | 1.38 (.573–3.316) | 0.474 |
| | No | 277 (92.3) | 291 (97) | Ref | | | |
| History of abortion | Yes | 55 (18.3) | 48 (16) | 0.449 | 1.18(.77–1.80) | NA | |
| | No | 245 (81.7) | 252 (84) | Ref | | | |
| History of surgical procedure | Yes | 31 (10.3) | 36 (12) | 0.517 | .845 (.508–1.41) | NA | |
| | No | 269 (89.7) | 264 (88) | Ref | | | |
| History of dental procedure | Yes | 42 (14) | 53 (17.7) | 0.220 | .759 (.488–1.18) | .642 (.397–1.038) | 0.070 |
| | No | 258 (86) | 247 (82.3) | Ref | | | |
| Body tattooing | Yes | 62 (20.7) | 34 (11.3) | 0.002 | 2.04 (1.295–3.21) | 1.66 (1.008–2.728) | 0.046 |
| | No | 238 (79.3) | 266 (88.7) | Ref | | | |
| Ear /nose piercings | Yes | 291 (97) | 287 (95.7) | 0.387 | 1.47 (.616–3.48) | NA | |
| | No | 9 (3) | 13 (4) | Ref | | | |
| History of home delivery | Yes | 14 (4.7) | 6 (2) | 0.077 | 2.4 (.909–6.33) | 1.61 (.550–4.72) | 0.384 |
| | No | 286 (95.3) | 294 (98) | Ref | | | |
| History of multiple sexual partners | Yes | 89 (29.7) | 39 (13) | 0.000 | 2.82(1.86–4.29) | 2.5(1.604–3.901) | 0.000 |
| | No | 211 (70.3) | 261 (87) | Ref | | | |
| female circumcision | Yes | 122 (40.7) | 110 (36.7) | 0.315 | 1.18 (.852–1.65) | NA | |
| | No | 178 (59.3) | 190 (63.3) | Ref | | | |
| History of hospital admission | Yes | 43 (14.3) | 33 (11) | 0.221 | 1.35 (.831–2.19) | 1.24 (.730–2.129) | 0.420 |
| | No | 257 (85.7) | 267 (89) | Ref | | | |
| Family history of HBV | Yes | 36 (12) | 12 (4) | 0.001 | 3.27 (1.67–6.42) | 2.62 (1.239–5.547) | 0.012 |
| | No | 264 (88) | 288 (96) | Ref | | | |
| Sharing sharp materials | Yes | 93 (31) | 32 (10.7) | 0.000 | 3.76 (2.42–5.85) | 3.02 (1.87–4.87) | 0.000 |
| | No | 207 (69) | 268 (89.3) | Ref | | | |
| Sharing tooth brushes | Yes | 12 (4) | 7 (2.3) | 0.249 | 1.74 (.677–4.49) | 1.32 (.464–3.775) | 0.601 |
| | No | 288 (96) | 293 (97.7) | | | | |
| History of jaundice | Yes | 18 (6) | 4 (1.3) | 0.005 | 4.72 (1.58–14.13) | 3.06 (.935–10.029) | 0.064 |
| | No | 282 (94) | 296 (98.7) | Ref | | | |
| History of contact with jaundice patient | Yes | 20 (6.7) | 13 (4.3) | 0.213 | 1.58 (.770–3.23) | .817 (.351–1.902) | 0.639 |
| | No | 280 (93.3) | 287 (95.7) | Ref | | | |
| Gravidity | primigravida | 118 (39.3) | 132 (44) | 0.247 | .825 (.596–1.142) | 1.05 (.733–1.505) | 0.789 |
| | Multigravida | 182 (60.7) | 168 (56) | Ref | | | |

CI: confidence interval, COR: Crude odds ratio, AOR: adjusted odds ratio, NA: not applicable refers to factors with p-value ≥0.25 in the bivariate analysis which was not considered in the Multivariable analysis

immunization status, methods of data collection, tests used to detect HBV infection, and the number of participants included in the study.

The socio-demographic characteristics of the study participants did not significantly differ in both the cases and the controls in this study. This is similar to studies in Ethiopia [22, 33] and Nigeria [34], indicating that socio-demographic characteristics were not risk factors for HBsAg positivity. On the contrary, other studies in Ethiopia [26, 35], Eritrea [19], Sudan [36], Cameroon [37], and Thailand [38] found a significant association between socio-demographic factors, such as marital status, age, occupation, religion, and HBV infection.

In our study body tattooing was significantly associated with HBV infection. This is inaccordance with studies from Ethiopia [27, 39–42], Cameron [37], and India [43]. The possible reason might be due to the use of unsterilized materials for the procedure because of low awererrnes on the transmission of HBV in the study site.But other reports from Ethiopia [24, 27], Nigeria [34], and Brazil [44] contradicted our findings. These differences might be due to variations in the level of awareness of the transmission of hepatitis viruses, safety precautions, and the culture of the society.

A history of having multiple sexual partners was another significant predictor of HBV infection in this study. Pregnant women with a history of multiple sexual partners were 2.5 (AOR 2.5 95% CI, 1.604–3.901) times more likely to develop HBV infection than their counterparts.The significant association of having multiple sexual partners with HBV infection was also reported by other studies from Gambella Hospital [26], Arba Minch Hospital [22], Felege Hiwot Referral Hospital [11], public hospitals of Wolaita Zone [7], hospitals in the Amhara rgegional state [41], Somalia [10], Cameron [37], East Africa [45], Nigeria [34], and India [43]. This might be due to sexual contact serving as a transmission route of HBV infection, which increases with the duration of sexual activity and the number of sexual partners [24].

The family history of HBV was also significantly associated with HBV infection. This is in line with similar studies reported from Ethiopia [5, 35, 41, 42], Egypt [6], Rwanda [46], and South Brazil [44]. This might be due to a lack of understuding about HBVtransmission methods and, the sharing of different personal and household items within the family, vertical transmission, and unsafe sexual practices.

Sharing sharp materials was also the other predictor variable associated with HBV infection. This agrees with studies in the Tigray region, Ethiopia [27], northern Palestine [47], and South Brazil [44]. Hepatitis B virus can survive and retain its infectivity for more than a week on the surface of non-sterile materials. So, the sharing of none sterile sharp materials due to a lack of awareness of the mode of transmission of HBV and other blood-borne infections might contribute to the transmission of HBV infection.

## Conclusions

The prevalence of HBV infection among pregnant women attending public hospitals in Addis Ababa was intermediate. No statistically significant association was observed between the acquisition of HBV infection and the socio-demographics. Body tattooing, history of having multiple sexual partners, family history of HBV, and sharing sharp materials were independent factors associated with HBV infection. Therefore awareness creation on the mode of transmission and early screening of all pregnant women for HBsAg must be strengthened to minimize and control the spread of the infection. Moreover, family members and sexual partners of HBV carriers should be screened and, if susceptible, receive the HBV vaccine and get awareness on HBV infection prevention.

## Limitation

Due to a lack of laboratory setups, the HBsAg status of the pregnant women was not confirmed using an enzyme-linked immunosorbent assay, and there might be false positive or false negative results. In addition, since the polymerase chain reaction test was not carried out in this study, pregnant women with occult HBV infections were not identified.

## Acknowledgments

We gratefully acknowledge the study participants, data collectors, and supervisors who participated in the data collection.

## Author Contributions

**Conceptualization:** Mebrihit Arefaine Tesfu, Tilahun Teklehaymanot Habtemariam, Nega Berhe Belay.

**Data curation:** Mebrihit Arefaine Tesfu.

**Formal analysis:** Mebrihit Arefaine Tesfu.

**Funding acquisition:** Tilahun Teklehaymanot Habtemariam.

**Investigation:** Mebrihit Arefaine Tesfu.

**Methodology:** Mebrihit Arefaine Tesfu, Tilahun Teklehaymanot Habtemariam, Nega Berhe Belay.

**Project administration:** Tilahun Teklehaymanot Habtemariam.

**Resources:** Nega Berhe Belay.

**Software:** Mebrihit Arefaine Tesfu.

**Supervision:** Mebrihit Arefaine Tesfu.

**Validation:** Mebrihit Arefaine Tesfu, Tilahun Teklehaymanot Habtemariam, Nega Berhe Belay.

**Visualization:** Mebrihit Arefaine Tesfu.

**Writing – original draft:** Mebrihit Arefaine Tesfu.

**Writing – review & editing:** Mebrihit Arefaine Tesfu, Tilahun Teklehaymanot Habtemariam, Nega Berhe Belay.

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
