## [Decision Letter · Decision Letter 0]

7 Dec 2022

PONE-D-22-22777RISK factors associated with Hepatitis B virus infection among pregnant women attending public hospitals in Addis Ababa, Ethiopia: A Case-control studyPLOS ONE

Dear Dr. Tesfu,

Thank you for submitting your manuscript to PLOS ONE. After careful consideration, we feel that it has merit but does not fully meet PLOS ONE’s publication criteria as it currently stands. Therefore, we invite you to submit a revised version of the manuscript that addresses the points raised during the review process.

Your manuscript was reviewed by 2 experts in the field. Both identified several important problems in your submission. Please review the attached comments and provide point-by-point responses.

We look forward to receiving your revised manuscript.

Kind regards,

Yury E Khudyakov, PhD

Academic Editor

PLOS ONE

Journal Requirements:

Reviewers' comments:

Reviewer's Responses to Questions

**Comments to the Author**

1. Is the manuscript technically sound, and do the data support the conclusions?

Reviewer #1: Yes

Reviewer #2: Yes

2. Has the statistical analysis been performed appropriately and rigorously? 

Reviewer #1: Yes

Reviewer #2: Yes

3. Have the authors made all data underlying the findings in their manuscript fully available?

Reviewer #1: Yes

Reviewer #2: Yes

4. Is the manuscript presented in an intelligible fashion and written in standard English?

Reviewer #1: No

Reviewer #2: Yes

5. Review Comments to the Author

Reviewer #1: The authors aimed to study the magnitude of HBV infection and its risk factors among pregnant women who attended public hospitals in Addis Ababa (Ethiopia). Although the issue is not new, they have included a large population in a place where infection is a heavy burden of disease, even when diagnosis is easy and infection can be prevented.

However, the study lacks several major aspects that severely impair the scientific significance and needs some changes:

- After reading the manuscript, it appears that the authors propose a multicentre prospective cohort study with a nested case-control study, although it is not clearly reflected in the Methods section.

- The tables are unclear, mainly table 2. I would consider simplifying it to allow a more comprehensible reading.

- Formatting of references does not follow the "Vancouver” style.

- The authors should revise the language to improve readability.

Reviewer #2: The research done by Tesfu et al. on risk factors associated with Hepatitis B virus infection among pregnant women has public health importance and screened large sample size about 12,138 pregnant women. So it good and in the range of PLOS One journal. However there are points must be corrected before publication.

1. RISk factor is written in capital letters, please correct it Risk factor.

2. Why do you exclude HIV infected patents, because sexual transmitted infection as one of risk factor in your study?

3. In line 125-126, correct in this way. Variables significant at p <0.25 with the dependent variable in bivariate analysis were selected for multivariable analysis.

4. In result part, in table one marital status and religion, it says others, you should have to specify below in the table as foot note.

5. I have seen your recommendation but you have not specified the stake, please try to make that.

Major comment

1. In the method part, sample size part, there is no sample size calculation. How do you reach 300 case and 300 control , it is not clear?

2. What is your sampling technique; it is not indicated in the method part? Is it quota sampling or random what?

3. You have used predesigned and pre-tested, where do you pre-test your questionnaire, it is not mentioned well?

4. The test used for the diagnosis of HBsAg is rapid test for the selection of cases and controls which is weak for this purpose , it would have been good to use ELISA or PCR?

6. PLOS authors have the option to publish the peer review history of their article (what does this mean?). If published, this will include your full peer review and any attached files.

Reviewer #1: No

Reviewer #2: No

---

## [Author Response · Author response to Decision Letter 0]

28 Feb 2023

Thank you for giving us the opportunity to submit a revised manuscript of<< Risk factors associated with Hepatitis B virus infection among pregnant women attending public hospitals in Addis Ababa, Ethiopia >>for publication in the journal of PLOS ONE. We appreciate the time and effort that you and the reviewers dedicated to provide feedback on our manuscript and are grateful for the insightful comments on and valuable improvements to our paper .We have incorporated almost all of the suggestions made by the reviewers. Those changes are marked within the manuscript. Please see below, in blue, for a point-by-point response to the reviewers’ comments.

 Academic editors and Reviewer’s comment to the authors 

1. RISk factor is written in capital letters, please correct it Risk factor.

Author response: We agree and corrected it.

2. Why do you exclude HIV infected patents, because sexual transmitted infection as one of risk factor in your study?

 Author response: Since HIV and HBV have similar route of transmission the risk factors may be over lapped that is why we exclude HIV seropositive pregnant woman. 

3. In line 125-126, correct in this way. Variables significant at p <0.25 with the dependent variable in bivariate analysis were selected for multivariable analysis.

Author response: we accepted the reviewers comment and corrected it.

4. In result part, in table one marital status and religion, it says others, you should have to specify below in the table as foot note.

Author response: we agree and corrected it. 

5. I have seen your recommendation but you have not specified the stake, please try to make that.

Author response: we agree and revised it.

Major comment

1. In the method part, samplesize part, there is no sample size calculation. How do you reach 300 case and 300 control, it is not clear?

Author response: we accept the comment and corrected it as follow 

To determine the MTCT of HBV and the protective effect of the immune-prophylaxis vaccine in Ethiopia, a sample size of 369 HBV-positive pregnant women was calculated using, 40% of MTCT of HBV in the absence of post-exposure prophylaxis in Nigeria, and giving any particular outcome to be with 5% marginal error, 95% confidence interval, and using the following single population determination formula. 

 n = (z ∂/2) 2 p (1-p)

 d2 

 To get 369 HBsAg positive, 12,318 pregnant women were screened for HBsAg as routine antenatal care (ANC) service in the selected public hospitals during the study period, and 300 of them were included in this study. The women whose screening results were positive for HBsAg were considered as cases. The purposive sampling technique was used to get the cases, and, for each HBsAg positive woman, the next HBsAg negative woman was interviewed and acted as a control. Thus 300 cases and 300 controls were involved in the study.

2. What is your sampling technique; it is not indicated in the method part? Is it quota sampling or random what?

Author response: The purposive sampling technique was used to get the cases, and, for each HBsAg positive woman, the next HBsAg negative woman was interviewed and acted as a control.

3. You have used predesigned and pre-tested, where do you pre-test your questionnaire, it is not mentioned well?

Author response: we accept the comment and revised it as follow

Questionnaires were carefully designed and pre-tested with individuals equivalent to 5% of the calculated sample size in Ras Desta Damtew Hospital and were slightly amended after pre-tested results revealed a lack of clarity for some responses.

4. The test used for the diagnosis of HBsAg is rapid test for the selection of cases and controls which is weak for this purpose; it would have been good to use ELISA or PCR?

Author response: we agree also with this comment .But we could not do those tests because they were not easily accessible and we describe this as a limitation of the study.

---

## [Editor Report · Decision Letter 1]

6 Mar 2023

PONE-D-22-22777R1Risk  factors associated with Hepatitis B virus infection among pregnant women attending public hospitals in Addis Ababa, EthiopiaPLOS ONE

Dear Dr. Tesfu,

Thank you for submitting your manuscript to PLOS ONE. After careful consideration, we feel that it has merit but does not fully meet PLOS ONE’s publication criteria as it currently stands. Therefore, we invite you to submit a revised version of the manuscript that addresses the points raised during the review process.

Thank you for submitting the revised version of your manuscript. You have responded to all comments by reviewers. However, the manuscript still needs revision.

Risk factor associated with HBV status among pregnant women are common for acquisition of HBV infections. The discussion section would be improved by comparing to risk factors in general population.

This is a research paper rather than policy and cannot contain recommendations.  The observations made in the study, however, may be used to indicate importance of the identified factors for enhancing screening for HBV infections among pregnant women. Since the study did not involve application of vaccines and immunoglobulins, it is not proper to conclude on their administration.  

It is not sufficient to mention the use of low sensitivity assays in Limitations. Potential effects of these assays on the presented findings should be spelled out.

We look forward to receiving your revised manuscript.

Kind regards,

Yury E Khudyakov, PhD

Academic Editor

PLOS ONE
---

## [Author Response · Author response to Decision Letter 1]

3 Apr 2023

Manuscript Number: PONE-D-22-22777R1

Manuscript title: Risk factors associated with Hepatitis B virus infection among pregnant women attending public hospitals in Addis Ababa, Ethiopia

Response to reviewers

Dear Dr. Yury E Khudyakov ( Acadamic editor of PLOS ONE journal)

Thank you for giving us the opportunity to submit a revised manuscript of<< Risk factors associated with Hepatitis B virus infection among pregnant women attending public hospitals in Addis Ababa, Ethiopia >>for publication in the journal of PLOS ONE. We appreciate the time and effort that you and the reviewers dedicated to provide feedback on our manuscript and are grateful for the insightful comments on and valuable improvements to our paper .We have incorporated all of the suggestions made by the reviewers. Those changes are marked within the manuscript. Please see below, in blue, for a point-by-point response to the reviewers’ comments.

 Academic editors and Reviewer’s comment to the authors 

1. Risk factors associated with HBV status among pregnant women are common for acquisition of HBV infections. The discussion section would be improved by comparing to risk factors in general population.

Author's response: We agree and we included and comparing the risk factors of HBV infection of the pregnant women with the general population.

2. This is a research paper rather than policy and cannot contain recommendations. The observations made in the study, however, may be used to indicate the importance of the identified factors for enhancing screening for HBV infections among pregnant women. Since the study did not involve the application of vaccines and immunoglobulins, it is not proper to conclude on their administration. 

Author's response: we agree and corrected it. 

3. It is not sufficient to mention the use of low sensitivity assays in Limitations. Potential effects of these essays on the presented findings should be spelled out.

Author's response: we agree and corrected as << Due to a lack of laboratory setups, the HBsAg status of the pregnant women was not confirmed using an enzyme-linked immunosorbent assay, and there might be false positive or false negative results. In addition, since the polymerase chain reaction test was not carried out in this study, pregnant women with occult HBV infections were not identified.>. 

Once more we thank you for the comments. We believe our manuscript is now ready for publishing. 

Sincerely,

Mebrihit Arefaine

On behalf of all authors.

---

## [Editor Report · Decision Letter 2]

5 Apr 2023

Risk  factors associated with Hepatitis B virus infection among pregnant women attending public hospitals in Addis Ababa, Ethiopia

PONE-D-22-22777R2

Dear Dr. Tesfu,

We’re pleased to inform you that your manuscript has been judged scientifically suitable for publication and will be formally accepted for publication once it meets all outstanding technical requirements.

Kind regards,

Yury E Khudyakov, PhD

Academic Editor

PLOS ONE